# Using MaxEnt to Predict the Potential Distribution of the Little Fire Ant (*Wasmannia auropunctata*) in China

**DOI:** 10.3390/insects13111008

**Published:** 2022-11-01

**Authors:** Mengfei Mao, Siqi Chen, Zengyuan Ke, Zengqiang Qian, Yijuan Xu

**Affiliations:** 1Red Imported Fire Ant Research Center, South China Agricultural University, Guangzhou 510642, China; 2College of Life Sciences, Shaanxi Normal University, 620 Western Chang’an Street, Xi’an 710119, China

**Keywords:** potential distribution, *Wasmannia auropunctata*, MaxEnt

## Abstract

**Simple Summary:**

In this study, based on known distribution points and climate variables, we used the MaxEnt model to predict and analyze the potential geographical distribution of *W. auropunctata* in China and study the relationship between the distribution of *W. auropunctata* and environmental variables. The performance level of the model was “good”. The results showed that most of the area south of the Yangtze River is potentially suitable for *W. auropunctata*, and temperature is the main factor affecting its distribution. The contemporary total suitable living area of *W. auropunctata* is 1,954,300 km^2^, accounting for 20.36% of China’s total land area. In future climate-change scenarios, the low-impact areas were widespread. Based on our results, we recommend that the government carry out *W. auropunctata* monitoring as soon as possible to determine the specific distribution and occurrence of *W. auropunctata* in the country.

**Abstract:**

Invasive ants are some of the most destructive species in ecosystems and can have serious ecological and socioeconomic impacts. The little fire ant, *Wasmannia auropunctata*, is native to Central and South America and was listed as one of the 100 most threatening major invasive organisms in the world by the International Union for Conservation of Nature (IUCN). The presence of *W. auropunctata* was first reported on the Chinese mainland in January 2022, but its distribution in China is still unclear. In this study, MaxEnt was used to predict the potential distribution of *W. auropunctata* in China based on known distribution points and climatic variables. The prediction results showed that most of the area south of the Yangtze River is potentially suitable for *W. auropunctata*, and temperature is the main factor affecting its distribution. The contemporary total suitable living area of *W. auropunctata* is 1,954,300 km^2^, accounting for 20.36% of China’s total land area. Further attention should be given to the potential impact of *W. auropunctata* invasions, and effective measures should be taken to eliminate the introduced population in China.

## 1. Introduction

With the development of geographic information systems (GISs) and computer statistical techniques, the potential geographical distribution of species can be inferred by analyzing climatic conditions in suitable areas [1]. A species distribution methodology uses algorithms to link the known distribution point data of the target species to related environmental variables, constructs a model to determine the ecological requirements required for species distribution, and projects the results of the operation to predict the potential distribution of species in a specific period and region in the future [2,3]. In recent years, a variety of species distribution models have emerged, such as BIOCLIM, ENFA, CART, MaxEnt, GAM, and GLM, and have been widely applied in the fields of ecology, conservation biology, and biogeography [4]. MaxEnt exhibits good stability; even if the distribution data information and environmental variables of the distribution area are incomplete, the potential distribution area of the species can be accurately predicted [5,6,7,8].

Invasive alien species (IASs) are one of the most important threats to native species, with serious ecological and socioeconomic impacts [9]. Together with other invasive ants, such as *Anoplolepis gracilipes*, *Solenopsis invicta*, *Linepithema humile*, and *Pheidole megacephala*, the little fire ant *Wasmannia auropunctata* was listed as one of the 100 most threatening major invasive species in the world by the International Union for Conservation of Nature (IUCN). *W. auropunctata* originated in Central and South America and is currently distributed in South America, North America, Australia, Italy, Israel, West Africa, the West Pacific Islands, and other regions [10,11]. *W. auropunctata* was found in the Wuri District, Taichung City, Taiwan Province, China, in 2021, and in Shantou City, Guangdong Province, in January 2022, which was the first time that the invasion of *W. auropunctata* was reported on the Chinese mainland [12,13]. In invasive areas, *W. auropunctata* is polygyne, exhibits supercolonial social organization, high interspecific aggression, strong thermal tolerance plasticity and adaptability, and has a huge impact on native species [11,14,15]. *W. auropunctata* lives in reciprocal symbiosis with invasive hemipteran insects (e.g., aphids), which can provide edible honeydew; the small fire ants provide protection for aphids, increasing the population of hemipteran pests and indirectly endangering agricultural production [16,17].

The growth and reproduction of plants and animals are closely related to climatic factors, and climate change may affect the pattern of biodiversity [18,19]. Global climate change makes a difference in temperature and precipitation patterns, and global warming and other climate-change processes will make its original habitat unable to provide stable living conditions, forcing species to migrate to other habitats with suitable climatic conditions for their survival and reproduction [20]. Models based on temporal and spatial aspects can establish monitoring procedures that act as early warning signals during climate change [21].

Predicting the future distribution of invasive species is critical to prioritization, early detection, and control. This study of the potential geographical distribution of *W. auropunctata* is expected to provide a reference for the prediction, forecasting prevention, and control of *W. auropunctata*.

## 2. Materials and Methods

### 2.1. Species Occurrence Data for W. auropunctata

Distribution data on *W. auropunctata* were obtained by retrieving data and information from the Global Biodiversity Information Facility (GBIF) (https://www.gbif.org, accessed on 7 July 2022) and the literature database (CNKI, Springer, ScienceDirect, and Web of Science). The latitude and longitude of distribution points without coordinate information were obtained from a Baidu map (https://api.map.baidu.com/lbsapi/getpoint/index.html, accessed on 7 July 2022). Then, the data statistics function of Excel was used to remove duplicate and widely marked distribution points, and the longitude and latitude data were converted to decimals.

Through the spatial analysis function of ArcGIS (version 10.8), redundant distribution point data were deleted using fishnet to reduce the impact of spatial autocorrelation [22], and 85 effective distribution points were obtained (Figure 1 and Appendix A). These distribution points had corresponding climate variables.

### 2.2. Environmental Variable Screening and Data Processing

We initially downloaded current conditions (1970–2000) and future climate data (2030s: 2021–2040, 2050s: 2041–2060, 2070s: 2061–2080, and 2090s: 2081–2100) and 19 bioclimatic variables (BIO01–BIO 19, 2.5 arc–min resolution, ~5 km) from the WorldClim website (https://www.worldclim.org/data/index.html, accessed on 16 July 2022). These variables are often used in many ecological and biogeographic studies for modeling species distribution. First, the 85 worldwide location data and 19 bioclimatic variables for *W. auropunctata* were imported into ArcGIS. The climate variables corresponding to the distribution points were extracted through the spatial analysis function of ArcGIS, and the Pearson correlation coefficient between the climate variables was calculated using SPSS (version 25) software (Appendix A). If the correlation between the two climate variables was greater than 0.8, only one climate variable was selected for use in the model. We converted them to the ASCII format by using ArcGIS as the Maxent layer for predicting *W. auropunctata* distributions. The climate variables together with the 85 distribution point data were imported into MaxEnt (version 3.4.4; https://biodiversityinformat-ics.amnh.org/open_source/maxent/, accessed on 18 July 2022) to produce preliminary models, and the initial percentage contribution, impermissibility importance, and jackknife analysis were calculated. Then, climate factors with very-low-percentage contributions were removed [23].

### 2.3. Species Distribution Model Establishment, Optimization, and Evaluation

In this study, we predicted the potential distribution of *W. auropunctata* assuming four different shared socioeconomic pathway scenarios (SSP1-2.6: Low forcing category, radiative forcing reaches 2.6 W/m^2^ in 2100; SSP2-4.5: Medium forcing category, radiative forcing reaches 4.5 W/m^2^ in 2100; SSP3-7.0: High forcing category, radiative forcing reaches 7.0 W/m^2^ in 2100; SSP5-8.5: High forcing category, radiative forcing reaches 8.5 W/m^2^ in 2100) for different periods (present, 2030s, 2050s, 2070s, and 2090s) [24].

We used the ENMeval package in R v3.6.1 to optimize the MaxEnt model and set the regulatory multiplier (RM) to 0.1–4, and each interval was 0.1, for a total of 40 regulatory multipliers. We used 15 feature classes (FCs): L, Q, P, H, LQ, LP, LH, QP, QH, PH, LQP, LQH, LPH, QPH, and LQPH. The model provides 5 features: Linear features (L), quadratic features (Q), product features (P), segmented features (H), and threshold features (T) [25,26]. The ENMeval data package was used to test the 600 parameter combinations, and we ultimately used the Akaike information criterion (AIC) model of the Akaike information criterion and used a 5% training omission rate (OR5) and the difference between the AUC values (AUCDIFF) to check the fit and complexity of the model [27]. By optimizing the model, we finally chose the characteristic class (FC) = LQ, regularization multiplier (RM) = 0.2, and ∆ AICc = 0 to build the best candidate model.

Based on these parameter settings, the selected distribution data, and environmental variables, we used MaxEnt to predict the modern potential distribution area, and the model was established and run 10 times. To ensure that the probability of the *W. auropunctata* distribution appeared close to reality, we selected 75% of the data for model training and the remaining data for model testing.

The proportion of test data was set as ‘Random seed’, the replicated run type was set as ‘Subsample’, the maximum iterations was set to 5000, the importance of climatic variables was measured by a ‘Jackknife test’, the impact of variables on the distribution of *W. auropunctata* was analyzed by creating response curves, the output format was logistic, and other settings were set to the default values in the software [28]. After the selection of the best model, the model results were transferred to the China region for four different shared socioeconomic pathway scenarios based on the 6th assessment report of the IPCC.

The receiver operating characteristic (ROC) curve and area under the ROC curve (AUC) were used to assess the accuracy of the MaxEnt predictions. The AUC is a threshold-independent metric that measures the models’ ability to distinguish between random and background points. The range of AUC was from 0 to 1, and the closer the value was to 1, the greater the probability of species presence; an AUC of less than 0.7 indicated poor reliability, 0.7-0.8 indicated fair reliability, 0.8-0.9 indicated good reliability, and 0.9-1 indicated excellent reliability [29,30,31].

### 2.4. Hierarchical Classification and Geospatial Analyses of Species Distribution

We used DIVA-GIS v7.5 (http://www.diva-gis.org/, accessed on 20 July 2022) to average the predictions that corresponded to different atmospheric circulation models in the same era and reclassified them for measurement of the geographical area of the species and then divided the *W. auropunctata* suitable region into four grades: The unsuitable region (0–0.1069); the low-suitability region (0.1069–0.4046); the moderately suitable region (0.4046–0.7023); and the highly suitable region (0.7023–1) [32]. We calculated the suitable areas and the rate of change in the suitable areas under four different shared socioeconomic pathway scenarios in present and future periods. The distribution area of the binary suitable region under the same socioeconomic pathway scenarios of the current and next four periods was superimposed by DIVA-GIS to obtain a low-impact area by taking the minimum value. Low-impact areas refer to areas where species are suitable for distribution in all ages, that is, they are relatively less affected by climate change [33].

## 3. Results

### 3.1. Analysis of the Accuracy of the Model

Based on 85 distribution points and 9 climate variables for *W. auropunctata*, the AUC values of the training data and test data of the initial model were 0.844 and 0.8864 (Figure 2), respectively, and the performance level of the model was “good”, indicating that the prediction model was accurate. The difference between the training set and the test set AUC values was 0.0425.

### 3.2. Selection of Key Variables in the Model

The Pearson correlation coefficient, initial percentage contribution, permutation importance, and importance of climatic variables in the distribution of *W. auropunctata* based on jackknife analysis were used to filter 19 climate variables. Ultimately, nine variables used to build the final model remained (Table 1): Mean diurnal range (BIO02), isothermality (BIO03), max temperature of the warmest month (BIO05), annual temperature range (BIO07), mean temperature of the wettest quarter (BIO08), precipitation in the wettest quarter (BIO16), precipitation in the driest quarter (BIO17), precipitation in the warmest quarter (BIO18), and precipitation in the coldest quarter (BIO19).

Temperature and precipitation are the main variables affecting the distribution of *W. auropunctata*, and the annual temperature range (BIO07) plays a crucial role. The environmental envelope test evaluated the impact of precipitation and temperature variables on the distribution of *W. auropunctata* using DIVA-GIS software (Appendix A). The generated graph indicated that the temperature variables show a greater contribution to the *W. auropunctata* distribution, as 85.9% of all records used in the model occur on the envelope, which has the expanded range of annual precipitation (Bio 12), in contrast to the narrow range of the annual mean temperature (Bio 1). Among all the components, the top three contributors of the final model were BIO07 (58.3%), BIO02 (12%), and BIO17 (7.3%). The top three permutation importance levels are BIO07 (53.9%), BIO05 (14.1%), and BIO03 (10.1%) (Table 1). When using only a single variable, the variable with the greatest gain to the model is still BIO07, followed by the mean diurnal range (BIO02) (Figure 3). The response curve between environmental variables and the probability of species presence reflects the relationship. As shown in Figure 4**,** with the increase in the annual temperature change range, the probability of the occurrence of *W. auropunctata* gradually decreases, and when the temperature change range is greater than 25 °C, the probability of occurrence is only 10%. When the mean diurnal range (BIO02) reaches 12 °C, the probability of *W. auropunctata* occurrence is only 30% (Figure 4).

In summary, the main factors affecting the geographical distribution range of *W. auropunctata* are the annual temperature range and the mean diurnal range.

### 3.3. Potential Distribution of W. auropunctata in China

The potential suitable distribution area of *W. auropunctata* obtained from MaxEnt is shown in Figure 5. The total suitable area based on present climatic conditions includes the provinces south of the Yangtze River, including Hainan Province, Guangdong Province, Guangxi Province, Chongqing Municipality, Yunnan Province, Guizhou Province, Hunan Province, Hubei Province, Fujian Province, Zhejiang Province, Jiangsu Province, Anhui Province, Taiwan Province, central, eastern and southern Sichuan, and a small part of southeastern Tibet (Figure 5). The moderately and highly suitable areas are distributed in Hainan Province, Taiwan Province, southern Guangdong Province, southern Guangxi Province, southwestern Yunnan Province, and southeastern Tibet (Figure 5). The unsuitable areas, the low-suitability area, the moderately suitable area, and the highly suitable area encompass 764.56 × 10^4^ km^2^, 164.86 × 10^4^ km^2^, 28.76 × 10^4^ km^2^, and 1.82 × 10^4^ km^2^, respectively. The total suitable area comprises 195.43 × 10^4^ km^2^, representing 20.36% of China’s total land area (Figure 5).

In addition, the results show that the predicted suitable area is highly consistent with the area currently occupied by *W. auropunctata* in the Americas, but in regions such as western Europe, sub-Saharan Africa, and Southeast Asia where the current *W. auropunctata* distribution area is sporadically distributed, it also has a wide range of potential suitable habitats (Appendix A).

In the context of climate change, we predicted the potential distribution of *W. auropunctata* under four different shared socioeconomic pathway scenarios (SSP1-2.6, SSP2-4.5, SSP3-7.0, and SSP5-8.5) for different future periods (2030s, 2050s, 2070s, and 2090s) and conducted a comparative analysis (Figure 6 and Figure 7). Based on the emission scenario of SSP1-2.6, the total and low-suitability areas were first reduced, and in the 2050s, both areas were the smallest and then continued to increase. Under the SSP2-4.5 scenario, the total suitable area reached a maximum in the 2050s and then decreased. Under the SSP5-8.5 and SSP3-7.0 scenarios, the forecast results showed that from the 2030s to the 2090s, the total suitable area of *W. auropunctata* in China shows an overall decreasing trend.

The results show that under the emission scenarios of SSP3-7.0-2090S, SSP5-8.5-2070s, and SSP5-8.5-2090S, compared with the current era, the total suitable area changes by more than 10%, decreasing by 225,300 km^2^, 239,000 km^2^, and 491,100 km^2^, respectively. Habitat loss was mainly concentrated in the western part of Sichuan Province, Hubei Province, Hunan Province, Anhui Province, Jiangsu Province, and north-central Zhejiang Province. Based on the emission scenario of SSP5-8.5-2090S, the habitats in Anhui and Jiangsu Provinces were almost completely depleted (Figure 6 and Figure 7). The highly suitable areas continued to increase based on the scenario of SSP1-2.6 (2030S: 16,600 km^2^; 2050S: 20,400 km^2^; 2070S: 22,500 km^2^; 2090S: 22,800 km^2^) (Figure 6a and Figure 7).

In other scenarios, the total suitable areas did not change much and remained large south of the Yangtze River (Figure 6 and Figure 7). For example, under the SSP1-2.6-2030s, SSP2-4.5-2030s, SSP2-4.5-2090s, SSP3-7.0-2030s, SSP3-7.0-2070s, and SSP5-8.5-2050S scenarios, the variation in the total suitable areas was less than 3% (Figure 6 and Figure 7).

### 3.4. Low Impact Area

The low-impact area (LIA) was obtained by superimposing the binary prediction map of the suitable region of different eras and identifying the completely overlapping section (Figure 8). This was a relatively small area affected by climate change. The low-impact area based on different shared socioeconomic paths varied greatly, representing 91.77%, 92.86%, 83.53%, and 69.43% of the contemporary suitable areas. Based on different scenarios, Hainan Province, Guangdong Province, Guangxi Province, Yunnan Province, Guizhou Province, Taiwan Province, and the southeastern region of Tibet were considered low-impact areas. Under the SSP5–8.5 scenario, the low-impact area decreased significantly (Figure 8).

## 4. Discussion

The construction of ecological niche models has been widely used in the study of invasion biology and conservation biology. The sampling range and sample size are the key factors that determine the reliability of the simulation results of the species distribution model. To improve the accuracy of the model, we called upon 600 feature combinations in the ENMeval packet in R software. The results showed that the AUC given FC = LQ and RM = 0.2 for the parameter ROC curve was 0.844, indicating that the simulation effect of the MaxEnt model of the potential geographical area of the *W. auropunctata* distribution was accurate and reliable. In similar previous studies, ENMeval packets were rarely used for model optimization, and even studies using ENMeval packets generally considered fewer combinations of parameters [31,34]. Therefore, after comparing 600 optimization combinations, the conclusions were relatively accurate.

*W. auropunctata* was first discovered in the Chinese mainland in January 2022, but its exact distribution in China is unclear. In this study, the potential area of the *W. auropunctata* distribution in China was predicted by combining climatic variables and distribution data through MaxEnt, and the results showed that the contemporary suitable habitat was mainly distributed in the provinces south of the Yangtze River; the northern boundary reached the central part of Jiangsu Province, the central part of Anhui Province, the northern part of Chongqing, and the northeast of Sichuan, with a total suitable living area of 195.43 × 10^4^ km^2^, accounting for 20.36% of China’s land area. The range of suitable habitats was narrower than that based on Federan’s predictions, which may be related to the choice of thresholds [35]. Carolina Coulin’s predictions of *W. auropunctata*’s habitat showed that it can survive north of South Latitude 40 and along the Mediterranean coast, and the greatest probability was between latitudes of 20 to 30 degrees north [15], which was consistent with the highly suitable region (Taiwan Province) we predicted. All these findings indicate that *W. auropunctata* has a wide range of suitable distribution areas.

Based on the Maxent model, four representative greenhouse gas emission scenarios were considered in this study. Under the high-emission scenarios (SSP 3-7.0 and SSP 5-8.5), projections show that from the 2030s to the 2090s, the total suitable area of *W. auropunctata* in China decreases, which is consistent with Bertelsmaier’s results that the potential distribution of most invading ants decreases with climate change [36]. Additionally, under SSP3-7.0 and SSP5-8.5, the relatively stable suitable area is greatly reduced compared with the low (SSP1-2.6) and medium (SSP2-4.5) emission scenarios (Figure 8). This suggests that a sustained increase in temperature may have a negative impact on the species [37].

The occurrence, reproduction, and spread of invasive insects are closely related to climatic conditions, and temperature is a major factor explaining the distribution of insect species [38]. In addition, other climatic conditions, such as precipitation and radiation, also have a significant impact on species distributions [39]. In this study, the relationship between key climate variables and the probability of occurrence of *W. auropunctata* was analyzed, and the corresponding response curve was obtained. The main factor affecting the geographical distribution range of *W. auropunctata* was the annual temperature range, and the response curve showed that the probability of occurrence of *W. auropunctata* gradually decreased with the increases in the annual temperature change range. Carolina Coulin’s research showed that after 10 days of low-temperature (15 °C) stress, little fire ant’s CTmin (critical thermal minimum) reached 4.2 °C; after 10 days of high-temperature (35 °C) stress, CTmax (thermal critical maximum) reached 43 °C, and its workers began to move at a temperature slightly higher than the temperature of CTmin. *W. auropunctata* is geographically distributed in areas where the hottest months have the highest temperatures well below their CTmax. As a result, the workers’ activity is largely influenced by temperature. This may be related to the fact that little fire ant nests are superficial nests, generally under dead branches and stones [11].

Niche models are often used to describe the basic niches of species distribution rather than the actual niches. The prediction capability of the model is excellent, but similar to other species distribution models, there are some inevitable limitations. Due to a variety of biological and abiotic factors, the actual ecological niche is usually smaller than the basic ecological niche [40]. In this study, we used MaxEnt to predict the potential distribution of *W. auropunctata* in China accounting for climate change. Although the prediction model result was “good”, it only took into consideration the climate factors that impact the distribution of *W. auropunctata*, and other biological factors (such as competitors) were not analyzed. In addition, factors such as soil type, topographic and geomorphological features, human interference, dispersal ability, and physiological characteristics also affect the geographical distribution of species [41,42,43,44]; these factors may bias the predictions.

Based on the results of this study, *W. auropunctata* has a vast suitable area in China. To prevent the invasion and spread of *W. auropunctata*, monitoring points should be set up in suitable areas according to the occurrence patterns and biological characteristics of *W. auropunctata*. In Hainan Province, Guangdong Province, Guangxi Province, Yunnan Province, Guizhou Province, Taiwan Province, and the southeastern region of Tibet and other low-impact areas that are less affected by climate change, it is recommended that the relevant governments carry out *W. auropunctata* monitoring as soon as possible to determine the specific distribution and occurrence of *W. auropunctata* in the country. Taking active and effective measures to prevent and control damage, reduce the economic losses caused by *W. auropunctata*, and respond to their presence may effectively slow the spread of the species, especially in regions of Taiwan and Shantou in Guangdong Province, where they have been discovered.

## Figures and Tables

**Figure 1 insects-13-01008-f001:**
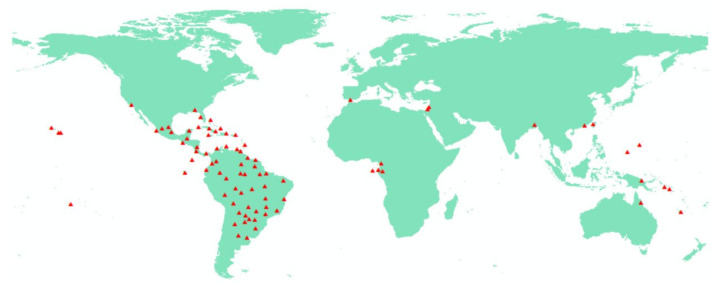
Species occurrence records on *W. auropunctata*. Notes: The red triangles indicate distribution points.

**Figure 2 insects-13-01008-f002:**
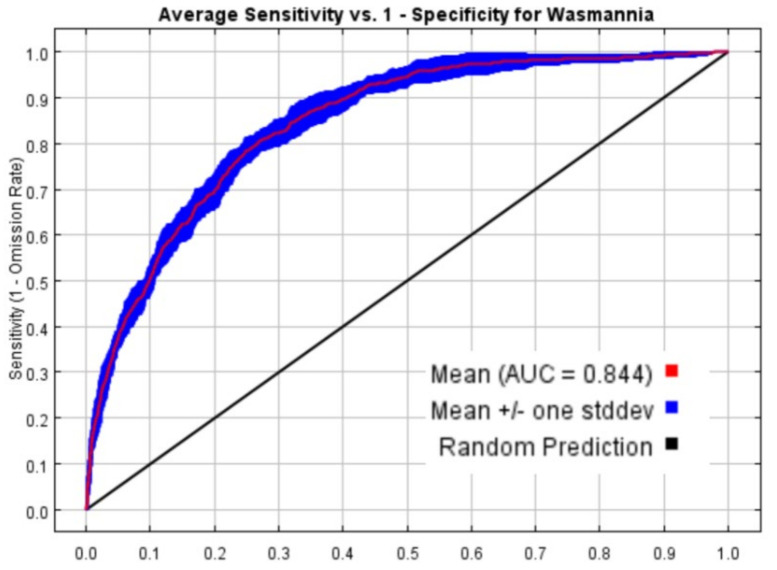
ROC curve and AUC values for the model. The average test AUC for the replicate runs was 0.844, and the standard deviation was 0.012.

**Figure 3 insects-13-01008-f003:**
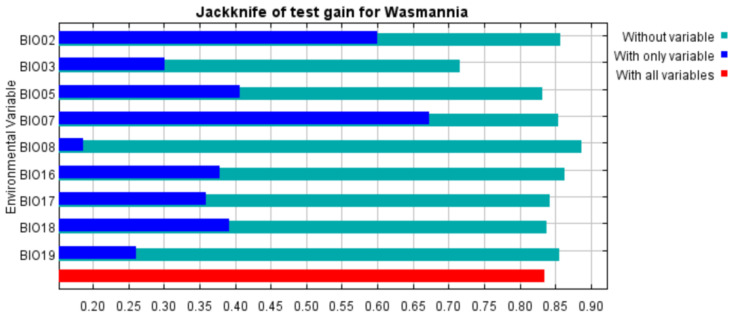
Jackknife test gain for *W. auropunctata*. Blue, green, and red bars represent running the MaxEnt model with the variable alone, without the variable, and with all variables, respectively.

**Figure 4 insects-13-01008-f004:**
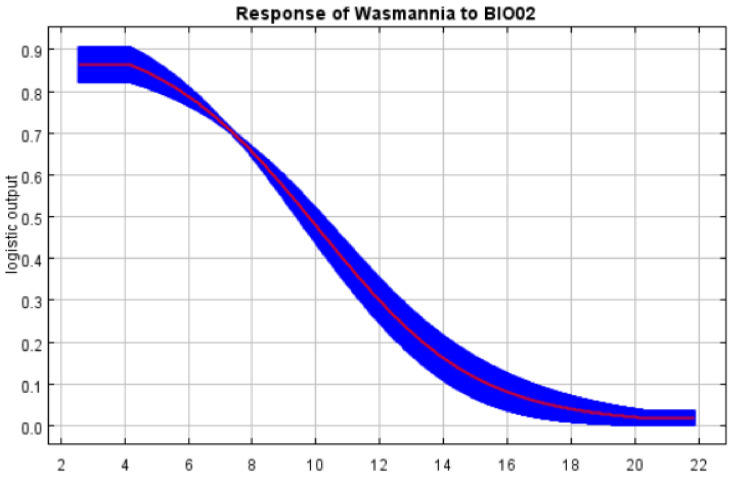
Response curve (BIO07, annual temperature range; BIO02, mean diurnal range).

**Figure 5 insects-13-01008-f005:**
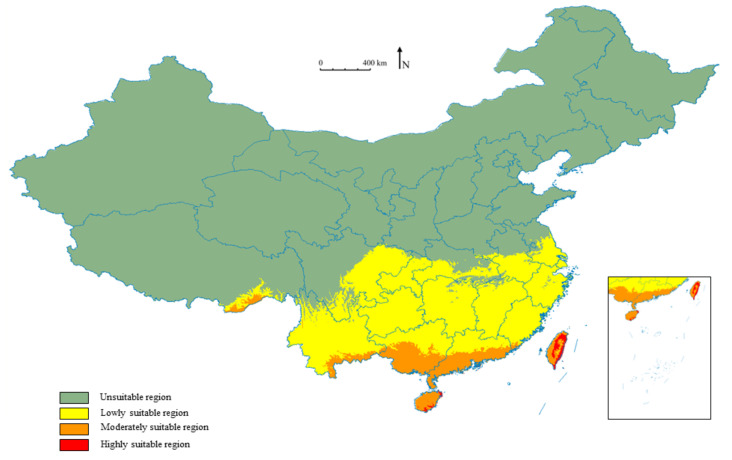
Potential distribution map of *W. auropunctata* in the current climate environment.

**Figure 6 insects-13-01008-f006:**
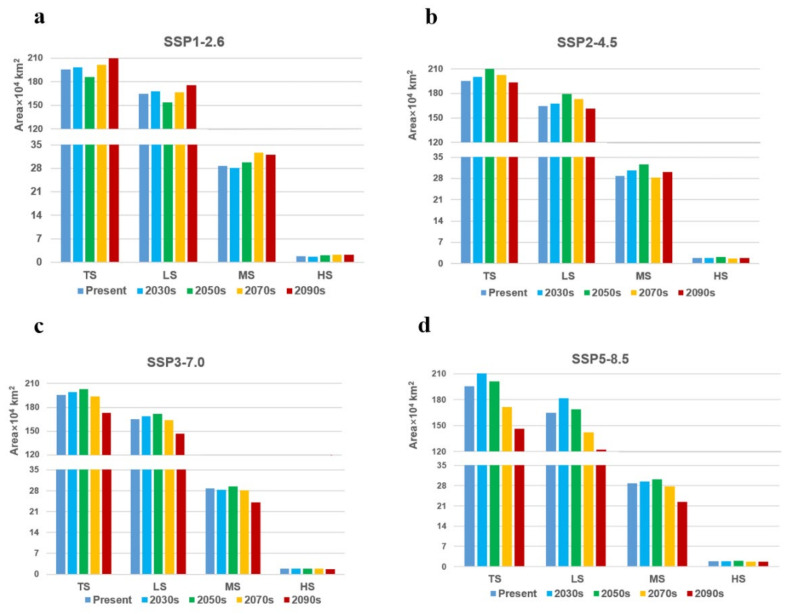
Present and future (2030s, 2050s, 2070s, and 2090s) suitable habitat area under four climate scenarios: (**a**) SSP1–2.6, (**b**) SSP2–4.5, (**c**) SSP3–7.0, and (**d**) SSP5–8.5. TS, LS, MS, and HS represent total suitable, low-suitability, moderately suitable, and highly suitable regions, respectively.

**Figure 7 insects-13-01008-f007:**
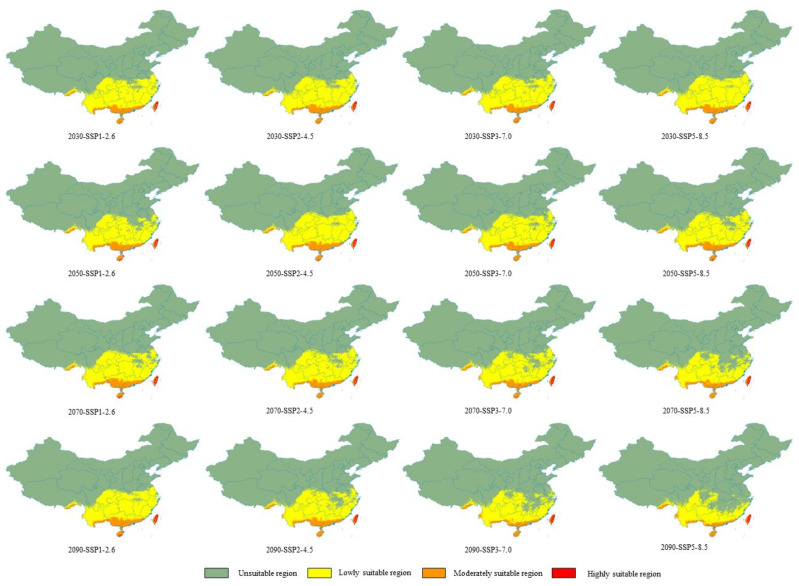
Maps of the *W. auropunctata* distribution in different climate scenarios and periods (2030s, 2050s, 2070s, and 2090s) of the 21st century. The shared socioeconomic pathways (SSPs) 1–2.6, SSP2–4.5, SSP3–7.0, and SSP5–8.5 represent four climate scenarios.

**Figure 8 insects-13-01008-f008:**
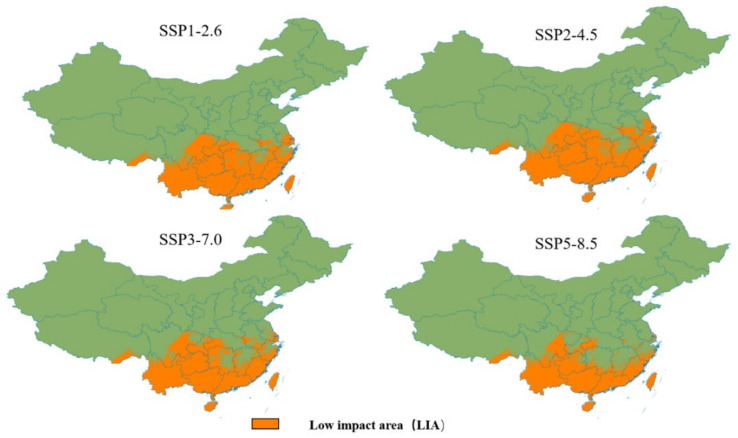
Prediction of low-impact areas (LIAs) for *W. auropunctata* under climate scenarios.

**Table 1 insects-13-01008-t001:** Climate variables and their contributions.

Variable	Environmental Variable	Percent Contribution	Permutation Importance
BIO07	Annual temperature range	58.8	53.9
BIO02	Mean diurnal range	12	2.2
BIO17	Precipitation of driest quarter	7.3	2.7
BIO18	Precipitation of warmest quarter	6.1	1.5
BIO03	Isothermality	5.5	10.1
BIO19	Precipitation of coldest quarter	4.5	4.9
BIO08	Mean temperature of wettest quarter	2.3	9.7
BIO05	Max temperature of warmest month	2.1	14.1
BIO16	Precipitation of wettest quarter	1.5	0.9

## Data Availability

The data presented in this study are available on request from the corresponding author.

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
