# Peer review of "Using MaxEnt to Predict the Potential Distribution of the Little Fire Ant (Wasmannia auropunctata) in China"

_insects, 2022, doi:10.3390/insects13111008_

Round 1

Reviewer 1 Report

The authors of the manuscript used MaxEnt to analyze the present potential distribution of the invasive ant Wasmannia auropunctata in China and predict its future potential distribution considering four different shared socioeconomic pathway scenarios (SSP1-2.6, SSP2-4.5, 104 SSP3-7.0, and SSP5-8.5) and analyzing 4 different periods (2030s, 2050s, 2070s and 2090s). The results are relevant since they constitute valuable information to aware about the potential spread of this invasive species in Asia.

Major comments:

1)     My major concern with this study is the omission of presence data in the native range that are key to predict the invasion of this species in temperate regions of the world (not only in tropical and subtropical regions). This distribution data are listed in Chifflet et al 2016, who carried out an extensive survey in the southern limit of native distribution of W. auropunctata in South America (including Argentina, Uruguay, the Atlantic Forest in Brazil, Bolivia and Paraguay). These points are not listed in the data base the authors used (GBIF), and I consider that including these presence data points may change the results of this study, because it is known the ability of W. auropunctata to survive in temperate regions, and that Bio 6 (minimum temperature of the coldest month) is one of the most important variables determining the species' distribution (see also Coulin et al 2018).  Bio 6 was not even considered in this manuscript to perform the SNM.

(see Chifflet, L., Rodriguero, M. S., Calcaterra, L. A., Rey, O., Dinghi, P. A., Baccaro, F. B., ... & Confalonieri, V. A. (2016). Evolutionary history of the little fire ant Wasmannia auropunctata before global invasion: inferring dispersal patterns, niche requirements and past and present distribution within its native range. Journal of evolutionary biology29(4), 790-809).

My recommendation is to add these presence data points and repeat the SDM (see Table S2 in
Chifflet et al, 2016).

2)     Also, the authors analyze the future potential distribution of W. auropunctata in China considering four different climate change scenarios but do not conclude anything about the differences in the results between the four scenarios. Thus, it is not clear what was the aim for performing those analyses. I would suggest they include in the introduction a statement about the importance of considering climate change scenarios in SDMs. Also add in M&M section what the four shared socioeconomic pathway scenarios chosen consist of. And finally, in the discussion section, the differences in the results between the four scenarios should be more discussed.

3)     Finally, I think English language and style should be revised.

Specific Comments:

Line 12-14: I suggest re-writing this sentence deleting “W. auropunctata has a wide range of distributions, and”.

Line 17: Delete “relevant”, replace “governments” with “government” and write “W. auropunctata” in italic.

Line 49: correct the species names, they are misspelled.

Line 54: cite also Wetterer and Porter (2003) for the species distribution.

Lines 57-59: Add citations for this sentence.

Lines 59-63: These sentences should be re-written, W. auropunctata is mentioned too many times.

Line 67: delete the first “and”.

Line 73-74: re-write this sentence (the word “literature” is repeated twice).

Line 104-105: Explain these 4 socioeconomic pathway scenarios and add citation.

Line 194: do you mean the total suitable area?

Paragraph in lines 193-205: I suggest verbs in present to talk about present climatic conditions

Line 194: I suggest replacing “was in” with “includes”.

Line 199: I suggest replacing “were” with “are”.

Line 203: I suggest replacing “were” with “encompass”.

Line 204: I suggest re-writing this sentence as: “The total suitable area comprise…”

Lines 214-226: Results here are not clearly presented. This paragraph could be improved.

Line 215-216: How is it possible that for the same socioeconomic pathway scenario (SSP5-8.5), the decrease in the total suitable area is bigger for 2070 than for 2090 compared to the current scenario?

Line 214-216: This result coincides with the study of Bertelsmaier et al (2014) (see Bertelsmeier, C., Luque, G. M., Hoffmann, B. D., & Courchamp, F. (2015). Worldwide ant invasions under climate change. Biodiversity and conservation24(1), 117-128). You should mention this study.

Line 223-224: Please indicate which result drives you to this statement because I do not appreciate this in the figures. In figure 6b the HS area for 2070 and 2090 seems to be the same.

Line 225-226: Specify for which scenarios the total suitable areas do not change.

Fig. 6: In figure 6a, for more clarity you should use the same colors as in the maps of figures 5 and 7. Otherwise you should delete figure 6a and incorporate this information in the plots 6 b, c, d and e. Moreover, why didn’t you include the total suitable area in figure 6a?

Fig. 6 b, c, d, e: As mentioned in my previous comment, you should include present in these plots so it is easier to visualize how the areas increase or decrease respect to present

Line 236: Correct the “)” position

Line 240-247: Explain better what a low impact area is. Please provide citations about this concept in the context of biological invasions. Also, you should have explained how you obtained this area in M&M section.

Line 256: This figure legend should be improved by specifying more information.

Line 261-262: That is why it is important to incorporate the distribution data in the southern limit of native distribution (surveys carried out by Chifflet et al 2016), which constitutes a subtropical/temperate region.

Line 290-295: This paragraph could be more discussed. Please propose a hypothesis that may explain the differences between the 4 SSP in the potential future distribution of W. auropunctata. Also, it is not clear to me what is the relevance of the low impact areas, what do these areas mean and why the authors give them such importance.

Line 290-291: This sentence is not clear.

Line 296-297: Temperature is very important but not the only factor explaining insect distribution. You should mention and cite other factors.

Line 310: by “few” do you mean “little”??

Line 312-313: There have been a lot of reviews on the limitations of ecological niche modeling. The authors should discuss some limitations to this approach and how they may influence their conclusions. For example, see Fitzpatrick, M. C., Weltzin, J. F., Sanders, N. J., & Dunn, R. R. (2007). The biogeography of prediction error: why does the introduced range of the fire ant over‐predict its native range?. Global Ecology and Biogeography16(1), 24-33.

Line 315-316: you could have used other abiotic factors such as disturbance, human land use or perform a mechanistic approach using CTmin as in Coulin et al 2018.

Line 316-319: the study of Rey et al (2012) suggests that this adaptation/evolution that you mention occurred before invasion (within the native range). You could have used this geographic data points and the bias of your study would have been smaller (See Chifflet et al 2016).

Author Response

Reviewer: 1

Comments and Suggestions for Authors

The authors of the manuscript used MaxEnt to analyze the present potential distribution of the invasive ant Wasmannia auropunctata in China and predict its future potential distribution considering four different shared socioeconomic pathway scenarios (SSP1-2.6, SSP2-4.5, 104 SSP3-7.0, and SSP5-8.5) and analyzing 4 different periods (2030s, 2050s, 2070s and 2090s). The results are relevant since they constitute valuable information to aware about the potential spread of this invasive species in Asia.

  • My major concern with this study is the omission of presence data in the native range that are key to predict the invasion of this species in temperate regions of the world (not only in tropical and subtropical regions). This distribution data are listed in Chifflet et al 2016, who carried out an extensive survey in the southern limit of native distribution of auropunctatain South America (including Argentina, Uruguay, the Atlantic Forest in Brazil, Bolivia and Paraguay). These points are not listed in the data base the authors used (GBIF), and I consider that including these presence data points may change the results of this study, because it is known the ability of W. auropunctata to survive in temperate regions, and that Bio 6 (minimum temperature of the coldest month) is one ofthe most important variables determining the species' distribution (see also Coulin et al 2018). Bio 6 was not even considered in this manuscript to perform the SNM.

Response: The range of sample collection will have a significant impact on the prediction results of the species distribution model. Generally speaking, the collected sample points have a uniform coverage and a wide area, and the richer the environmental information obtained, the more accurate the prediction of the potential distribution areas of species will be. Therefore, we first collected distribution points including Argentina, Uruguay, the Atlantic Forest in Brazil, Bolivia and Paraguay, and then filtered the distribution point data through ArcGIS to make it more evenly distributed and improve the accuracy of model predictions, without using all the distribution points in the region, only a small part. BIo06 is discharged at modeling time because its contribution to the initial percentage of the model is extremely low.

(see Chifflet, L., Rodriguero, M. S., Calcaterra, L. A., Rey, O., Dinghi, P. A., Baccaro, F. B., ... & Confalonieri, V. A. (2016). Evolutionary history of the little fire ant W. auropunctata before global invasion: inferring dispersal patterns, niche requirements and past and present distribution within its native range. Journal of evolutionary biology, 29(4), 790-809).

My recommendation is to add these presence data points and repeat the SDM (see Table S2 in
Chifflet et al, 2016).

  • Also, the authors analyze the future potential distribution of auropunctata in China considering four different climate change scenarios but do not conclude anything about the differences in the results between the four scenarios. Thus, it is not clear what was the aim for performing those analyses. I would suggest they include in the introduction a statement about the importance of considering climate change scenarios in SDMs. Also add in M&M section what the four shared socioeconomic pathway scenarios chosen consist of. And finally, in the discussion section, the differences in the results between the four scenarios should be more discussed.

Response: Global climate change makes a difference in temperature and precipita-tion patterns, which will cause changes in the original habitat conditions of species, and the distribution areas of most species will also change. Generally speaking, Based on the emission scenario of SSP1-2.6, the total suitable area increases; On the other hand, under the high emission scenario (SSP3-7.0 and SSP5-8.5), the total suitable area decreased. SSP2-4.5 have no obvious changes.(See Line 66-74  115-118  242-248  338-346)

3)Finally, I think English language and style should be revised.

Specific Comments:

Line 12-14: I suggest re-writing this sentence deleting “W. auropunctata has a wide range of distributions, and”.

Response: Had modified. (See Line 12-14)

Line 17: Delete “relevant”, replace “governments” with “government” and write “W. auropunctata” in italic.

Response: Had modified. (See Line 17)

Line 49: correct the species names, they are misspelled.

Response: Had modified. (See Line 59-60)

Line 54: cite also Wetterer and Porter (2003) for the species distribution.

Lines 57-59: Add citations for this sentence.

Response: Had modified. (See Line 61)

Lines 59-63: These sentences should be re-written, W. auropunctata is mentioned too many times.

Response: Had modified. (See Line 60-63)

Line 67: delete the first “and”.

Response: Had modified

Line 73-74: re-write this sentence (the word “literature” is repeated twice).

Response: Had modified(See Line 83-84)

Line 104-105: Explain these 4 socioeconomic pathway scenarios and add citation.

Response: SSP1-2.6: Low forcing category, radiative forcing reaches 2.6 W/m2 in 2100; SSP2-4.5: Medium forcing category, radiative forcing reaches 4.5 W/m2 in 2100;SSP3-7.0: High forcing category, radiative forcing reaches 7.0 W/m2 in 2100; SSP5-8.5: High forcing category,radiative forcing reaches 8.5 W/m2 in 2100. (See Line 115-119)

Line 194: do you mean the total suitable area?

Response:Yes. (See Line 218)

Paragraph in lines 193-205: I suggest verbs in present to talk about present climatic conditions

Line 194: I suggest replacing “was in” with “includes”.

Line 199: I suggest replacing “were” with “are”.

Line 203: I suggest replacing “were” with “encompass”.

Line 204: I suggest re-writing this sentence as: “The total suitable area comprise…”

Response: Thank you for advice. (See Line 217-229)

Lines 214-226: Results here are not clearly presented. This paragraph could be improved.

Response: Had modified(See Line 242-264)

Line 215-216: How is it possible that for the same socioeconomic pathway scenario (SSP5-8.5), the decrease in the total suitable area is bigger for 2070 than for 2090 compared to the current scenario?

Response: Sorry, there is a mistake here. (See Line 249-251 and Table S3)

Line 214-216: This result coincides with the study of Bertelsmaier et al (2014) (see Bertelsmeier, C., Luque, G. M., Hoffmann, B. D., & Courchamp, F. (2015). Worldwide ant invasions under climate change. Biodiversity and conservation, 24(1), 117-128). You should mention this study.

Response: Added. (See Line 346)

Line 223-224: Please indicate which result drives you to this statement because I do not appreciate this in the figures. In figure 6b the HS area for 2070 and 2090 seems to be the same.

Response:2070S: 22500 km2; 2090S:22800 km2  There are subtle differences between them.

Line 225-226: Specify for which scenarios the total suitable areas do not change.

Response: Had modified. (See Line 265-268)

Fig. 6: In figure 6a, for more clarity you should use the same colors as in the maps of figures 5 and 7. Otherwise you should delete figure 6a and incorporate this information in the plots 6 b, c, d and e. Moreover, why didn’t you include the total suitable area in figure 6a?

Fig. 6 b, c, d, e: As mentioned in my previous comment, you should include present in these plots so it is easier to visualize how the areas increase or decrease respect to present

Response: It has been redrawn. (See Fig. 6) Table S3 is added, which will be clearer.

Line 236: Correct the “)” position

Response: Had modified. (See Line 277)

Line 240-247: Explain better what a low impact area is. Please provide citations about this concept in the context of biological invasions. Also, you should have explained how you obtained this area in M&M section.

Response: I'm sorry, this noun has only been found in Chinese explanation. (See Line 158-162)

Line 256: This figure legend should be improved by specifying more information.

Response: Had modified. (See Line 301)

Line 261-262: That is why it is important to incorporate the distribution data in the southern limit of native distribution (surveys carried out by Chifflet et al 2016), which constitutes a subtropical/temperate region.

Response: In order to make the distribution of data points more uniform, we only use part of the data in this area.

Line 290-295: This paragraph could be more discussed. Please propose a hypothesis that may explain the differences between the 4 SSP in the potential future distribution of W. auropunctata. Also, it is not clear to me what is the relevance of the low impact areas, what do these areas mean and why the authors give them such importance.

Response: The distribution area of binary suitable region under the same socioeconomic pathway scenarios of the current and the next four years was superimposed by DIVA-GIS to ob-tain a low impact area by taking the minimum value. Low-impact areas can show the impact of climate change on species distribution more obviously.

Line 290-291: This sentence is not clear.

Response:Sorry, it has been rewritten.

Line 296-297: Temperature is very important but not the only factor explaining insect distribution. You should mention and cite other factors.

Response: Had modified. (See Line 352-353  378-380)

Line 310: by “few” do you mean “little”??

Response: Had modified. (See Line 365)

Line 312-313: There have been a lot of reviews on the limitations of ecological niche modeling. The authors should discuss some limitations to this approach and how they may influence their conclusions. For example, see Fitzpatrick, M. C., Weltzin, J. F., Sanders, N. J., & Dunn, R. R. (2007). The biogeography of prediction error: why does the introduced range of the fire ant over‐predict its native range?. Global Ecology and Biogeography, 16(1), 24-33.

Response: Had modified. (See Line 368-377)

Line 315-316: you could have used other abiotic factors such as disturbance, human land use or perform a mechanistic approach using CTmin as in Coulin et al 2018.

Line 316-319: the study of Rey et al (2012) suggests that this adaptation/evolution that you mention occurred before invasion (within the native range). You could have used this geographic data points and the bias of your study would have been smaller (See Chifflet et al 2016).

Response: Thank you for suggestion.

Reviewer 2 Report

The manuscript “Using MaxEnt to predict the potential distribution of the little fire ant (Wasmannia auropunctata) in China” uses MaxEnt to create species distribution models for the likely invadable range of Wasmannia aurapunctata, which was recently introduced to mainland China. The authors demonstrate that much of the southern portion of the country could likely be suitable habitat for this destructive invasive. The core of the project is interesting but would benefit from an expanded perspective of the entire region rather than focusing on a single country. In addition the manuscript itself needs significant editing to provide additional clarity in the methods and to remove redundancy in the discussion. I have outlined these issues and other minor considerations in detail below:

1. W. aurapunctata is a globally important invasive species that, with its recent established introduction into eastern Asia is likely to disrupt ecosystem service and agricultural practices throughout the region. W. aurapunctata will not limit its spread based on national boundaries and as such, limiting the exploration of its potential invadable range to only China artificially limits the value of this project. Invasive species do not recognize national boundaries and repeated invasions are likely if consideration of the entire invadable range is not made. In addition, W. aurapunctata is not likely to spread to island habitats without human intervention. It would make much more sense to focus on spread across contiguous landmasses. I would strongly suggest reconsidering models to examine the potential range of W. aurapunctata in a greater region beyond China that is more biologically relevant to the focal species.

2.  As currently written the methods and justification for model selection are particularly opaque and difficult to follow. I am not confident I could replicate the results given the information provided. Cited literature including Muscarella et al 2014 and Steen et al 2019 and other similar studies using MaxEnt to model ant ranges (see citations below) provide very clear examples of species distribution models and their methods for variable selection, data pruning, etc. The manuscript would benefit from following these examples as a framework for how best to present the methods of a MaxEnt model.

Gull E. Fareen A, Mahmood T, Bodlah I, Rashid A, Khalid A, Mahmood S (2022) Modeling potential distribution of newly recorded ant, Brachyponera nigrita using Maxent under climate change in Pothwar region, Pakistan. PLoS ONE 17(1): e0262451. https://doi.org/10.1371/journal.pone.0262451

Fitzpatrick, M. C., N. J. Gotelli, and A. M. Ellison. 2013. MaxEnt versus MaxLike: empirical comparisons with ant species distributions. Ecosphere 4(5):55.

http://dx.doi.org/10.1890/ES13-00066.1

3. The discussion should be simplified to focus only on the implications of the results in relationship to larger theory or similar work. The repetition of results within the discussion are unnecessary. Specifically the 1st and 3rd paragraphs (starting at lines 260 and 290, respectively) could be entirely removed as they only repeat information presented in the results and do not tie this information to larger processes or information. The 2nd and 4th paragraphs present novel connections of this work with other literature and are very interesting but could also benefit from some additional pruning of redundant results presentations.

Minor issues

1. There are small grammatical mistakes throughout the manuscript. None of these impede interpretability but I have pointed out some examples out to highlight some things to look for.

Line 16 – drop “the” from “In the future climate change scenarios” as “scenarios” is plural

Line 17 – change “the relevant governments” to “the relevant government agencies”

Line 30 – change “Great” to “More” or “Further”
Line 230 – change “€” to “(e)”

2. Line 55 – The original citation for the first arrival of Wasmannia auropunctata in Taiwan should be cited here, especially given that it is a very recent publication:

Lee et al 2021 - First Record of the Invasive Little Fire Ant (Wasmannia auropunctata) (Hymenoptera: Formicidae) in Taiwan: Invasion Status, Colony Structure, and Potential Threats

Author Response

Reviewer: 2

Comments and Suggestions for Authors

The manuscript “Using MaxEnt to predict the potential distribution of the little fire ant (Wasmannia auropunctata) in China” uses MaxEnt to create species distribution models for the likely invadable range of Wasmannia aurapunctata, which was recently introduced to mainland China. The authors demonstrate that much of the southern portion of the country could likely be suitable habitat for this destructive invasive. The core of the project is interesting but would benefit from an expanded perspective of the entire region rather than focusing on a single country. In addition the manuscript itself needs significant editing to provide additional clarity in the methods and to remove redundancy in the discussion. I have outlined these issues and other minor considerations in detail below:

1.W. aurapunctata is a globally important invasive species that, with its recent established introduction into eastern Asia is likely to disrupt ecosystem service and agricultural practices throughout the region. W. aurapunctata will not limit its spread based on national boundaries and as such, limiting the exploration of its potential invadable range to only China artificially limits the value of this project. Invasive species do not recognize national boundaries and repeated invasions are likely if consideration of the entire invadable range is not made. In addition, W. aurapunctata is not likely to spread to island habitats without human intervention. It would make much more sense to focus on spread across contiguous landmasses. I would strongly suggest reconsidering models to examine the potential range of W. aurapunctata in a greater region beyond China that is more biologically relevant to the focal species.
Response:It supplements the potential distribution map of W. aurapunctata in the world under the current climate environment (See Line 230-234 and Fig. S2).

2.As currently written the methods and justification for model selection are particularly opaque and difficult to follow. I am not confident I could replicate the results given the information provided. Cited literature including Muscarella et al 2014 and Steen et al 2019 and other similar studies using MaxEnt to model ant ranges (see citations below) provide very clear examples of species distribution models and their methods for variable selection, data pruning, etc. The manuscript would benefit from following these examples as a framework for how best to present the methods of a MaxEnt model.
Response: The data of species distribution points(Table S1)and Pearson correlation coefficient(Table S2) are added, which is helpful to understand this research. (See Line 111-112). The third reviewer said “methodology written well and can easily be replicated.”

 Gull E. Fareen A, Mahmood T, Bodlah I, Rashid A, Khalid A, Mahmood S (2022) Modeling potential distribution of newly recorded ant, Brachyponera nigrita using Maxent under climate change in Pothwar region, Pakistan. PLoS ONE 17(1): e0262451. https://doi.org/10.1371/journal.pone.0262451

Fitzpatrick, M. C., N. J. Gotelli, and A. M. Ellison. 2013. MaxEnt versus MaxLike: empirical comparisons with ant species distributions. Ecosphere 4(5):55.

http://dx.doi.org/10.1890/ES13-00066.1

  1. The discussion should be simplified to focus only on the implications of the results in relationship to larger theory or similar work. The repetition of results within the discussion are unnecessary. Specifically the 1st and 3rd paragraphs (starting at lines 260 and 290, respectively) could be entirely removed as they only repeat information presented in the results and do not tie this information to larger processes or information. The 2nd and 4th paragraphs present novel connections of this work with other literature and are very interesting but could also benefit from some additional pruning of redundant results presentations.
    Response: Delete parts of these two paragraphs and rephrase them. (See Line and 333-346 and 309-313).
    Minor issues

    1. There are small grammatical mistakes throughout the manuscript. None of these impede interpretability but I have pointed out some examples out to highlight some things to look for.

Line 16 – drop “the” from “In the future climate change scenarios” as “scenarios” is plural

Response: Had modified. (See Line 16)

Line 17 – change “the relevant governments” to “the relevant government agencies”

Response: Had modified. (See Line 17)

Line 30 – change “Great” to “More” or “Further”
Line 230 – change “€” to “(e)”
Response: Had modified. (See Line 30 and 272)
2. Line 55 – The original citation for the first arrival of Wasmannia auropunctata in Taiwan should be cited here, especially given that it is a very recent publication:

Lee et al 2021 - First Record of the Invasive Little Fire Ant (Wasmannia auropunctata) (Hymenoptera: Formicidae) in Taiwan: Invasion Status, Colony Structure, and Potential Threats

Response: Thank you for suggestion. (See Line 58)

Reviewer 3 Report

It was interesting to me to revise the work entitled "sing MaxEnt to predict the potential distribution of the little fire ant (Wasmannia auropunctata) in China" the study looks comprehensive and effective for this important species. I have no critical comments on the work and methodology written well and can easily be replicated. My only comments and suggestions:

1. Provide the coordinates of the records you used in this study as a supplementary file.

2. As you used DIVA-GIS please add enveloped test between Bio01 and Bio12 to the methodology section or as supplementary  this test will indicate how the species records used, behave in two-dimensional niche analysis as in:

Hosni, E.M., Nasser, M.G., Al-Ashaal, S.A. et al. Modeling current and future global distribution of Chrysomya bezziana under changing climate. Sci Rep 10, 4947 (2020). https://doi.org/10.1038/s41598-020-61962-8

Finally, I appreciate your efforts and work. It sure could be published in Insects 

Author Response

Reviewer: 3

Comments and Suggestions for Authors

It was interesting to me to revise the work entitled "sing MaxEnt to predict the potential distribution of the little fire ant (Wasmannia auropunctata) in China" the study looks comprehensive and effective for this important species. I have no critical comments on the work and methodology written well and can easily be replicated. My only comments and suggestions:

1.Provide the coordinates of the records you used in this study as a supplementary file.

Response: Thank you for suggestion. (Table S1)

  1. As you used DIVA-GIS please add enveloped test between Bio01 and Bio12 to the methodology section or as supplementary this test will indicate how the species records used, behave in two-dimensional niche analysis as in:

Hosni, E.M., Nasser, M.G., Al-Ashaal, S.A.et al.Modeling current and future global distribution of Chrysomya bezziana under changing climate. Sci Rep 10, 4947 (2020). https://doi.org/10.1038/s41598-020-61962-8

Response: Added to supplementary materials. (Fig S1 and line 188-194)

Finally, I appreciate your efforts and work. It sure could be published in Insects

Round 2

Reviewer 1 Report

All comments have been addressed. I have some minor comments to improve the ms:

L.10-12: You should mention here why did you used maxent (the aim): “In this study, based on the known distribution points and climate variables, we chose the MaxEnt model to…”

L. 11: replace “chose” with “used”.

L. 11-12: Delete “and called 600 feature combinations in the ENMeval packet 11 in R software” (this is too much M&M detail for the simple summary)

L. 12: Delete “prediction”

L. 17: I suggest rewrite this sentence as: “Based on our results we recommend the government carry out…”

L. 49: Delete “Buren”

L. 58: Replace “are” with “is”

L. 58-61: I suggest unifying these two sentences as: “In invasive areas W. auropunctata is polygyne, exhibits supercolonial social organization, high interspecific aggression, strong thermal tolerance plasticity and adaptability, and has a huge impact on native species.

L. 66-74: This paragraph is not well written, you should improve it

L. 77: Delete the first “and”

L. 82: Delete “First,”

L. 89: Delete “Second,”

L. 89-90: Rewrite as “redundant distribution point data were deleted…”

L. 109-112: Rewrite as: “The climate variables together with the 85 distribution point data were imported into MaxEnt (version 3.4.4; https://biodiversityinformat-ics.amnh.org/open_source/maxent/) to calculate the initial percentage contribution. Then, climate factors with very low percentage contribution were removed [23].

L. 159: by “years” do you mean “periods”? please correct it

L. 189: a space is missing in “variables on”

L. 191: delete the “g” in gmore

L. 230-234: I do not understand what the authors mean in this paragraph

L. 230: show, instead of  “shows”

L. 230: is, instead of “are”

L 242-243: Relpace with “Based on the emission scenario of SSP1-2.6the total and low suitable areas were first reduced…”

L. 245: Replace “area of total suitable region” with “the total suitable area”

L. 261-263: This should be mentioned above, whe you talk about the SSP1 (Line 242-244)

L. 289-290: You have already said this in M&M, no need to repeat

L. 343: Rewrite as: “Under the high emission scenarios (SSP 3-7.0 and SSP 5-8.5)

L. 345: Replace “shows a decreasing trend overall” with “decreases”

L. 346-349: Rewrite as: “Also under SSP3-7.0 and SSP5-8.5, the relatively stable suitable area is greatly reduced compared with the low (SSP1-2.6) and medium (SSP2-4.5) emission scenarios (fig. 8). This suggests that a sustained increase in temperature may have a negative impact on the species.

L. 372-373: Shoud write “…, only considering the climate factors that impact the distribution of W. auropunctata.”

L. 375-378: Delete this sentence, it is not well written

L. 379-380: Rewrite as: “own dispersal ability and physiological characteristics also affect…” and add here the citation 41 ([41-44])

Author Response

All comments have been addressed. I have some minor comments to improve the ms:

L.10-12: You should mention here why did you used maxent (the aim): “In this study, based on the known distribution points and climate variables, we chose the MaxEnt model to…”

Authors’ response: Modified.

  1. 11: replace “chose” with “used”.

Authors’ response: Modified.

  1. 11-12: Delete “and called 600 feature combinations in the ENMeval packet 11 in R software” (this is too much M&M detail for the simple summary)

Authors’ response: Deleted.

  1. 12: Delete “prediction”

Authors’ response: Modified.

  1. 17: I suggest rewrite this sentence as: “Based on our results we recommend the government carry out…”

Authors’ response: Modified.

  1. 49: Delete “Buren”

Authors’ response: Modified.

  1. 58: Replace “are” with “is”

Authors’ response: Modified.

  1. 58-61: I suggest unifying these two sentences as: “In invasive areas W. auropunctata is polygyne, exhibits supercolonial social organization, high interspecific aggression, strong thermal tolerance plasticity and adaptability, and has a huge impact on native species.

Authors’ response: Modified.

  1. 66-74: This paragraph is not well written, you should improve it

Authors’ response: Modified.

  1. 77: Delete the first “and”

Authors’ response: Modified.

  1. 82: Delete “First,”

Authors’ response: Modified.

  1. 89: Delete “Second,”

Authors’ response: Modified.

  1. 89-90: Rewrite as “redundant distribution point data were deleted…”

Authors’ response: Modified.

  1. 109-112: Rewrite as: “The climate variables together with the 85 distribution point data were imported into MaxEnt (version 3.4.4; https://biodiversityinformat-ics.amnh.org/open_source/maxent/) to calculate the initial percentage contribution. Then, climate factors with very low percentage contribution were removed [23].

Authors’ response: Modified.

  1. 159: by “years” do you mean “periods”? please correct it

Authors’ response: Modified.

  1. 189: a space is missing in “variables on”

Authors’ response: Modified.

  1. 191: delete the “g” in gmore

Authors’ response: Modified.

  1. 230-234: I do not understand what the authors mean in this paragraph

Authors’ response: According to the request of another reviewer, these two sentences are a simple description of the results of the global prediction of the potential suitable area of W. Auropunctata.

  1. 230: show, instead of “shows”

Authors’ response: Modified.

  1. 230: is, instead of “are”

Authors; response: Modified.

L 242-243: Relpace with “Based on the emission scenario of SSP1-2.6,the total and low suitable areas were first reduced…”

Authors’ response: Modified.

  1. 245: Replace “area of total suitable region” with “the total suitable area”

Authors’ response: Modified.

  1. 261-263: This should be mentioned above, whe you talk about the SSP1 (Line 242-244)

Authors’ response: Modified.

  1. 289-290: You have already said this in M&M, no need to repeat

Authors’ response: Modified.

  1. 343: Rewrite as: “Under the high emission scenarios (SSP 3-7.0 and SSP 5-8.5)

Authors’ response: Modified.

  1. 345: Replace “shows a decreasing trend overall” with “decreases”

Authors’ response: Modified.

  1. 346-349: Rewrite as: “Also under SSP3-7.0 and SSP5-8.5, the relatively stable suitable area is greatly reduced compared with the low (SSP1-2.6) and medium (SSP2-4.5) emission scenarios (fig. 8). This suggests that a sustained increase in temperature may have a negative impact on the species.

Authors’ response: Modified.

  1. 372-373: Shoud write “…, only considering the climate factors that impact the distribution of W. auropunctata.”

Authors’ response: Modified.

  1. 375-378: Delete this sentence, it is not well written

Authors’ response: Modified.

  1. 379-380: Rewrite as: “own dispersal ability and physiological characteristics also affect…” and add here the citation 41 ([41-44])

Authors’ response: Modified.

Thank you very much for taking the time to review this manuscript. We apologize for these grammar problems and have corrected it according to your suggestions. In addition, we have asked a native speaker for language editing.

Reviewer 2 Report

This is my second time reviewing the manuscript "Using MaxEnt to predict the potential distribution of the little fire ant (Wasmannia auropunctata) in China". The authors have improved the clarity of the discussion by focusing on the implications of the results and addressed a few of the grammatical mistakes I directly pointed out which does improve the overall flow of the paper. These changes have made for a much more poignant and interesting discussion.

However two of the three major issue I originally had with this manuscript remain and were given at best only cursory responses. More extensive consideration is needed. I have outlined these issues again below:

1. W. auropunctata is a globally important invasive species that has and will continue to have zero regard for national boarders. This paper focused on a recent introduction to a region that is likely to provide substantial new territory for W. auropunctata loses much of its utility by focusing on a national boarder rather than taking a more biotic perspective. As a modeling study, there is no reason to limit predictions to a single country when a regional perspective would be more informative and more biologically relevant.

Furthermore, by focusing on political boarders rather than climatically and biologically relevant regions, several islands are included that are 1) already invaded by W. auropunctata or 2) unlikely to experience invasion without direct human intervention given the motility of this ant. A biologically relevant focus would be more directed towards contiguous landmasses that ants are more likely to disperse across by their own means. 

2. MaxEnt models are ubiquitous in the literature and provide a sound means for assessing potential range shifts and changes following new invasions or under various climatic scenarios. There are many examples of streamlined and clear methods in other manuscripts, specifically some cited in this manuscript and in other manuscripts more closely aligned with the ant focused work presented here. The addition of the supplementary tables is helpful but the reproducibility of the results presented here hinge on the clarity of the methods. I would still suggest more closely aligning the presentation with that of other published literature.

Minor issues

1. The authors directly addressed the grammatical issues I presented in the first review but these were only a few examples of things to look for throughout the manuscript. There remain many small grammatical errors throughout the manuscript, including several in the newly edited text. None of these impede interpretability but should be fixed in any finalized version of this manuscript.

Author Response

This is my second time reviewing the manuscript "Using MaxEnt to predict the potential distribution of the little fire ant (Wasmannia auropunctata) in China". The authors have improved the clarity of the discussion by focusing on the implications of the results and addressed a few of the grammatical mistakes I directly pointed out which does improve the overall flow of the paper. These changes have made for a much more poignant and interesting discussion.

However two of the three major issue I originally had with this manuscript remain and were given at best only cursory Authors responses. More extensive consideration is needed. I have outlined these issues again below:

  1. W. auropunctata is a globally important invasive species that has and will continue to have zero regard for national boarders. This paper focused on a recent introduction to a region that is likely to provide substantial new territory for W. auropunctata loses much of its utility by focusing on a national boarder rather than taking a more biotic perspective. As a modeling study, there is no reason to limit predictions to a single country when a regional perspective would be more informative and more biologically relevant.

Furthermore, by focusing on political boarders rather than climatically and biologically relevant regions, several islands are included that are 1) already invaded by W. auropunctata or 2) unlikely to experience invasion without direct human intervention given the motility of this ant. A biologically relevant focus would be more directed towards contiguous landmasses that ants are more likely to disperse across by their own means.

Authors’ response: We agree with the reviewer's comments and have also added a potential distribution map of little fire ants around the world (Fig. S2). However, we hope the reviewers can understand that this research is funded by the government, and we are required to analyze the potential distribution of this newly invaded species in China, which will help guide the investigation of the current distribution of little fire ants in China.

  1. MaxEnt models are ubiquitous in the literature and provide a sound means for assessing potential range shifts and changes following new invasions or under various climatic scenarios. There are many examples of streamlined and clear methods in other manuscripts, specifically some cited in this manuscript and in other manuscripts more closely aligned with the ant focused work presented here. The addition of the supplementary tables is helpful but the reproducibility of the results presented here hinge on the clarity of the methods. I would still suggest more closely aligning the presentation with that of other published literature.

Authors’ response: We list the main steps, including the analysis methods of the MaxEnt models: 1. The acquisition and redundancy deletion of the longitude and latitude of the known distribution points of the species (lines 79-89); 2. acquisition of environmental variables and calculation of Pearson correlation between variables (lines 97-105); 3. the environmental variables and the screened distribution point data were imported into Maxent for model construction, and then the key environmental variables were screened (lines 105-108); 4. further model optimization (lines 121-136); and 5. model evaluation (lines 137-143). We have detailed the methods in the revised manuscript.

Thank you very much for providing the related literature, which we have cited in the Materials and Methods section. In the revised manuscript, we have tried to make the method clearer and more accurate. However, unlike the article you provided, we put the final result of environmental variable selection in section 3.2 (Selection of Key Variables in the Model).

Minor issues

  1. The authors directly addressed the grammatical issues I presented in the first review but these were only a few examples of things to look for throughout the manuscript. There remain many small grammatical errors throughout the manuscript, including several in the newly edited text. None of these impede interpretability but should be fixed in any finalized version of this manuscript.

Authors’ response: We apologize for the poor language of our manuscript. In addition, we have asked a native speaker for language editing.